# Robust Normalization of Luciferase Reporter Data

**DOI:** 10.3390/mps2030062

**Published:** 2019-07-25

**Authors:** Andrea Repele

**Affiliations:** Department of Biology, University of North Dakota, Grand Forks, ND 58202, USA

**Keywords:** luciferase reporter, transfection efficiency, normalization, gene regulation, promoter, enhancer

## Abstract

Transient Luciferase reporter assays are widely used in the study of gene regulation and intracellular cell signaling. In order to control for sample-to-sample variation in luminescence arising from variability in transfection efficiency and other sources, an internal control reporter is co-transfected with the experimental reporter. The luminescence of the experimental reporter is normalized against the control by taking the ratio of the two. Here we show that this method of normalization, “ratiometric”, performs poorly when the transfection efficiency is low and leads to biased estimates of relative activity. We propose an alternative methodology based on linear regression that is much better suited for the normalization of reporter data, especially when transfection efficiency is low. We compare the ratiometric method against three regression methods on both simulated and empirical data. Our results suggest that robust errors-in-variables (REIV) regression performs the best in normalizing Luciferase reporter data. We have made the R code for Luciferase data normalization using REIV available on GitHub.

## 1. Introduction

Transient reporter assays are an important and widely used tool in the study of gene regulation [1,2,3,4], intracellular cell signaling [5,6,7], and other areas of molecular, cellular, and developmental biology [8,9,10]. In a reporter assay, the activity of a promoter, potentially in combination with an enhancer, is measured by placing an inert reporter gene such as *luciferase* or *lacZ* under the control of the promoter. *luciferase* is commonly used as the reporter gene since the assay has both high sensitivity and a very high dynamic range [11].

An important technical challenge in the analysis of Luciferase reporter data is that transfection efficiency can vary significantly from sample to sample, especially in hard to transfect cell types such as primary cells. Other sources of random experimental error, such as the amount of transfected DNA, number of cells assayed, cell viability, and pipetting errors further compound transfection efficiency variation to yield Luciferase luminescence data that can vary over an order of magnitude [4,8,12]. For example, a reporter of *CCAAT/Enhancer binding protein, α* (*Cebpa*) promoter expression exhibits ∼3-fold variation in luminescence when assayed in PUER cells [13,14,15], a myeloid cell line with low transfection efficiency (Figure 1; Section 3.1).

The prevalent method of correcting for sample-to-sample luminescence variation [3,8] is to co-transfect an independent control reporter, such as *Renilla luciferase* expressed from a constitutive promoter, along with the promoter/enhancer reporter being assayed. Since transfection efficiency and other experimental errors would affect both constructs equally, the activity of the experimental promoter relative to the constitutive promoter can be determined by taking the ratio of firefly and *Renilla* luminescence. Let there be i=1,⋯,N samples and $F_{i}$ and $R_{i}$ represent firefly and *Renilla* luminescence in the *i*th replicate. Then, the relative activity is estimated as
(1)$$A=\frac{1}{N}\sum_{i=1}^{N}\frac{F_{i}}{R_{i}}.$$

Although it is in common use, “ratiometric” normalization is statistically unsound since it weights low- and high-luminescence replicates equally even though the former produce less reliable estimates of normalized reporter activity. This problem is especially acute in cell types having low transfection efficiency and, consequently, high variability in luminescence measurements.

Here we propose an alternative approach to the normalization of Luciferase reporter data that is more robust than the ratiometric method. Firefly luminescence is expected to be proportional to *Renilla* luminescence,
(2)F=AR,
and therefore, the relative activity *A* can be estimated as the slope of the best fit line using linear regression (Figure 1). Ordinary least-squares finds the best fit line by minimizing the errors in the dependent variable (*F*), that is, the distances between the line and the data points in the *vertical* direction. Ordinary least-squares (OLS) regression places a higher weight on high-luminescence points and thus avoids the main weakness of ratiometric normalization.

We also considered two alternatives to OLS to find the method best suited for the normalization of Luciferase data. OLS assumes that the values of the independent variable, *Renilla* luminescence in our case, are known exactly and do not include random errors. Since *Renilla* luminescence is itself a random variable in transient assays, we evaluated errors-in-variables (EIV) regression (Section 3.4.3; [16]). In contrast to OLS, which minimizes the vertical error, EIV regression minimizes the *perpendicular* distance of each data point from the line, leading to the minimization of errors in both the independent and the dependent variables. Lastly, we considered robust errors-in-variables (REIV) regression (Section 3.4.4; [17]). Luciferase luminescence data contain potential outliers (Figure 1; [4,8]), which can unduly influence activity estimates. In REIV, the estimation of the slope and intercept is rendered insensitive to outliers by utilizing a bounded loss function. This means that the contribution of any point to the error stops growing with distance if it lies far enough from the line. Thus, outliers, points that lie far away from the rest, are prevented from having an undue influence on the slope and intercept of the best fit line.

In this study we compared the four methods described above, ratiometric, OLS, EIV, and REIV, in two ways. First, we applied each method to simulated data and assessed its ability to recover the “true” activity used to generate the synthetic data. Second, we tested the methods on empirical measurements of *Cebpa* promoter and enhancer activity in PUER cells under two different cytokine treatments. Our results show that while ratiometric normalization performs poorly on high variability data from low transfection efficiency experiments and can lead to erroneous biological conclusions, REIV performs the best and robustly estimates activity under a variety of different conditions.

## 2. Results and Discussion

We evaluated the performance of four methods, ratiometric, OLS, EIV, and REIV, on simulated data, where the “ground truth” is known. Reporter assays in cells with low transfection efficiency yield activity data with a high coefficient of variation, making normalization more challenging. Other sources of experimental error impact normalization similarly, and accurate normalization becomes more challenging with increasing standard deviation of the experimental error. We aimed to evaluate the methods’ ability to recover the known promoter activity at different levels of transfection efficiency and experimental error.

We simulated sample-to-sample variation in transfection efficiency (*t*) (Section 3.5) by sampling from the Beta distribution. The coefficient of variation of the Beta distribution is inversely related to mean transfection efficiency [16] and consequently synthetic data from low transfection efficiency simulations have greater variability, mirroring experimental reality. Mean transfection efficiency ($\bar{t}$) was varied to assess how the methods perform as sample-to-sample variability increases. Random experimental errors, modeled with a Gaussian mixture model [17], were introduced into both *Renilla* and firefly simulated data (Section 3.5). The Gaussian mixture model includes low-frequency contamination by errors with a higher standard deviation and allowed us to simulate outliers (Figure 1). We varied the standard deviation of *Renilla* errors ($\sigma_{11}$) to assess the performance of each method at different levels of experimental variability. Robustness of each method to outliers was assessed by varying the standard deviation of the contaminating errors ($\sigma_{12}$). We also varied the sample size (*N*) and “true” activity level to investigate how they impact normalization.

Simulated data were generated for each combination of activity, transfection efficiency, sample size, and error standard deviations. Each normalization method was used to estimate activity from the exact same set of simulated data. The simulations were repeated M=300 times and the performance of the methods was assessed using three different measures. Let the known or true activity be *A* and the activity estimated in the *k*th simulation be $A_{k}$. We utilized the relative absolute difference between the true and estimated values,
(3)$$\frac{|A-A_{k}|}{A},$$
as a measure of the relative bias in the estimation. Second, we computed the relative median absolute deviation (MAD) of the estimated activities $A_{k}$ as a measure of precision. Although readily interpretable, activity based measures are less reliable when the true activity is high. Since the activity is the slope of the regression line, small deviations of the regression line from the true line lead to large deviations in activity when the true activity is large. For this reason, we also utilized a measure developed by Zamar (1989) [17],
(4)$$\frac{1}{M}\sum_{k=1}^{M}\left\{1-\frac{|1+A_{k}A|}{{\left(1+A_{k}^{2}\right)}^{\frac{1}{2}}{\left(1+A^{2}\right)}^{\frac{1}{2}}}\right\}.$$

The “Zamar criterion” is independent of the activity level since it is invariant under rotations of the xy-plane [17] and serves well to compare different methods. The Zamar criterion takes values between 0 and 1, the former implying perfect recovery of the true activity.

In the first set of simulations, we assessed the ability of the methods to infer the activity from simulated data lacking outliers ($\sigma_{12}=\sigma_{11}$) at different levels of mean transfection efficiency. The performance of all methods improved with increasing transfection efficiency (Figure 2a), however the regression-based methods, OLS, EIV, and REIV, performed better at lower transfection efficiencies as compared to the ratiometric method. At transfection efficiencies less than 25%, the ratiometric method produces rather inaccurate estimates of activity, and the 90th percentile of relative bias is ∼300% at 10% transfection efficiency. Next, we investigated whether the accuracy of the estimates could be improved by increasing sample size (*N*). Since the main difference between the ratiometric and the other methods arises at low transfection efficiencies, we set the mean transfection efficiency to 25% for all subsequent simulations. Whereas the performance of the regression based methods improved with increasing sample size, the bias in the activity inferred by the ratiometric method did not decrease with sample size (Figure 2b). These set of simulations suggested that the ratiometric method of normalization results in biased activity estimates that do not improve even at very large sample sizes.

In the third set of simulations, we investigated the effect of the activity level on the normalization (Figure 2c). The performance of all the methods deteriorated at activity levels close to 1. An activity level of 1 implies that firefly luminescence is comparable to *Renilla* luminescence, which only occurs for exceptionally weak promoters since it is common practice to transfect the firefly plasmid in a 10- to 200-fold excess over the *Renilla* plasmid. Next we investigated the effect of experimental error in *Renilla* luminescence (Figure 3a). As one would expect, the performance of all methods deteriorates as the magnitude of experimental error is increased. Furthermore, the ratiometric method had the worst performance amongst all the methods. The errors-in-variables methods, EIV and REIV, were the most resilient to errors in *Renilla* (Figure 3a; Zamar criterion) since OLS does not attempt to minimize errors in the independent variable. Finally, we investigated the performance of the methods in the presence of outliers. So far, the simulations had been carried out without outliers since the standard deviation of the contaminating errors was the same as the Renilla experimental error ($\sigma_{12}=\sigma_{11}$). We increased the standard deviation of the contaminating errors ($\sigma_{12}$) to model outliers. REIV had the best performance (Figure 3b; Zamar criterion) of all the methods in the presence of outliers. These simulations suggest that errors-in-variables methods, especially REIV, perform best when normalizing transient transfection data.

Having investigated the performance of the four methods on simulated data, we sought to test them on empirical data. We utilized a dataset of 85 measurements of firefly luminescence driven by the *Cebpa* promoter paired to *Renilla* measurements driven by the CMV promoter, acquired in 8 different experiments carried out over a 6 month period (Figure 4a). The Luciferase assays were performed in PUER cells (Section 3.1), which have a transfection efficiency of ∼30%. The data show that despite the variability in firefly and *Renilla* luminescence, they are linear with respect to each other. We sampled the *Cebpa* promoter luminescence dataset 1000 times with replacement at sample sizes N=3,6,10,20,30. Figure 4b shows boxplots of the normalized activity. The three regression based methods converge to a median activity of ∼13 as sample size is increased. Mirroring the results of the simulations, the ratiometric method estimates activity as ∼16 and this bias is not ameliorated by increasing the sample size. Suspecting that this bias arises from low-luminescence measurements, we sampled a subset of the data with *Renilla* luminescence greater than 8 relative luminescence units (RLU). The activity inferred by the regression based methods was unchanged as compared to the larger dataset but those inferred by the ratiometric method now agreed with the other three methods (Figure 4c). This result implies that while ratiometric estimation can be biased by low-luminescence measurements, the regression based methods are robust to such data.

The bias in the ratiometric estimates of activity can result in erroneous biological conclusions. We illustrate this with an example from our analysis of the regulation of *Cebpa* enhancers [4,15]. We tested the functional role of binding sites of Gfi1 and C/EBP family transcription factors (TFs) in an enhancer (Figure 5a) found ∼7 kb downstream of the *Cebpa* transcription start site [4]. We assayed the activity of three reporter vectors in PUER cells cultured in either IL3 without 4-hydroxy-tamoxifen (OHT) or in GCSF 24 h after induction with OHT. OHT induction in the presence of GCSF causes PUER cells to differentiate into neutrophils over a 7-day period [4,14]. The activity of the CMV promoter was not affected by cytokine treatment in PUER cells (Appendix A; Welch two sample *t*-test, N=10, p=0.93), allowing for a valid comparison between treatments. *Cebpa(0)* contained the proximal promoter, *Cebpa(7)* contained the enhancer in addition to the proximal promoter, and *Cebpa(7m1)* contained the enhancer in which the C/EBP sites had been mutated. Analysis with REIV (Figure 5b; left panels) shows that the enhancer upregulates promoter activity 4–6 fold in both conditions and that mutating the C/EBP sites reduces activity by about 50% in IL3 conditions, which is consistent with the known role of C/EBP TFs as activators [18] of *Cebpa*. In the data analyzed using REIV, mutating the C/EBP sites does not have an effect in GCSF conditions. Analysis of the same exact data with the ratiometric method however detects a statistically significant upregulation of *Cebpa(7m1)* in GCSF conditions (Figure 5b; right panels). The ratiometric method would thus lead one to the incorrect conclusion that C/EBP TFs repress *Cebpa* in GCSF conditions, which is inconsistent with their well-known role as activators.

Our results suggest that the commonly used ratiometric method performs adequately on reporter data from cells having transfection efficiency of at least 75% but can provide inaccurate estimates in low transfection efficiency experiments. The regression-based methods in contrast provide accurate inference irrespective of transfection efficiency. Our simulations show that OLS performs poorly when random errors in *Renilla* luminescence are large (Figure 3a) and both OLS and EIV perform poorly when the data have large outliers (Figure 3b). REIV was the most robust to all of these potential problems. In our tests on reporter data from PUER cells (Figure 4), the three regression based methods, OLS, EIV, and REIV, gave nearly identical estimates. This implies that the particular data used here neither have large random errors in *Renilla* luminescence nor a significant outlier problem. It cannot be assumed, however, that all datasets are free of these problems. The safest choice therefore would be to favor REIV over the other methods. The main cost of REIV is that it is more computationally intensive than the other methods. This is not a problem in practice, since REIV takes ∼30 s on modern desktop hardware. Use of REIV regression for normalization could enable studies in hard-to-transfect cell lines that were not possible before. We have made R code to analyze reporter assay data using EIV and REIV available on GitHub (Section 3.4.3 and Section 3.4.4). A detailed protocol for using the code to analyze dual-Luciferase reporter data is provided in Section 3.6.

## 3. Materials and Methods

### 3.1. Cell Culture

We utilized PUER cells, *Spi1*${}^{-/-}$ cells expressing conditionally activable PU.1 protein, to assay reporter activity [3,13,14]. PUER cells can be differentiated into neutrophils by PU.1 activation in the presence of Granulocyte Colony Stimulating Factor (GCSF). PUER cells were routinely maintained in complete Iscove’s Modified Dulbecco’s Glutamax medium (IMDM; Gibco, 12440061) supplemented with 10% FBS, 50 μM β-mercaptoethanol, 5 ng/mL IL3 (Peprotech, 213-13). Cells were differentiated into neutrophils by replacing IL3 with 10 ng/mL GCSF (Peprotech, 300-23) and inducing with 100 nM OHT after 48 h.

### 3.2. Construction of the Luciferase Reporter Plasmid

The *Cebpa* promoter [4,15] was cloned into a pGL4.10*luc2* Luciferase reporter vector (Promega, E6651) between the XhoI and HindIII sites. The enhancers were inserted between BamHI and SalI sites downstream of the SV40 late poly(A) signal. The promoter was amplified from genomic DNA of C57BL/6J mice with primers TGG CCT AAC TGG CCG GTA CCT GAG CTC GCT AGC CTC GAG AAC TCC TAC CCA CAG CCG CG (Fwd) and TCC ATG GTG GCT TTA CCA ACA GTA CCG GAT TGC CAA GCT TCA GCT TCG GGT CGC GAA TG (Rev), which include 40 bp of sequence homologous to pGL4.10*luc2*. PCR amplification was carried out using Q5 High-Fidelity 2X Master Mix (NEB, M0492L) following the manufacturer’s instructions. The following PCR cycling conditions were used: initial denaturation of 30 s at 98 ${}^{\circ }$C, 30 cycles of 30 s at 98 ${}^{\circ }$C, 30 s at 60 ${}^{\circ }$C, and 60 s at 72 ${}^{\circ }$C, and a final extension for 10 min at 72 C. Gibson Assembly (GA) reactions [19] were carried out using 0.06 pmol of digested vector and 0.18 pmol of insert, for 60 min at 50 ${}^{\circ }$C. NEB high-efficiency competent cells (NEB, E5510S) were transformed according to manufacturer’s instructions. See Repele et al. [4] for the construction of the reporter vectors having the wildtype and mutant enhancers.

### 3.3. Transfection and Luciferase Assays

PUER cells were transfected with *Cebpa* promoter reporter vector and *Renilla* control vector pRL-CMV (Promega, E2261) in a 1:200 ratio using a 4D-Nucleofector (Lonza). Cells were transfected with 2.26 μg total plasmid DNA in SF buffer (Lonza, V4SC-2096), using program CM134 and incubated for 24 h prior to luminescence measurement. Enhancer/promoter activity was assayed in the neutrophil differentiation by placing cells in medium containing GCSF and OHT after nucleofection. After incubation, firefly and *Renilla* luminescence were measured using the Dual-Glo Luciferase activity kit (Promega, E2920) and the DTX 880 Multimode Detector (Beckman Coulter) according to manufacturer’s instructions. Transfections were performed in at least 10 replicates.

### 3.4. Tested Normalization Methods

#### 3.4.1. Ratio

For each measurement, the ratio of firefly and *Renilla* luminescence was computed and averaged over the samples (Equation (Equation 1)).

#### 3.4.2. Ordinary Least-Squares Regression

The normalized activity of the construct was determined as the slope of the line F=AR, where *F* and *R* are firefly and *Renilla* luminescence respectively, using ordinary least-squares (OLS) regression. The regression was performed using the lm function of R.

#### 3.4.3. Errors-in-Variable Regression

The normalized activity of the construct was determined as the slope of the line F=AR using errors-in-variables (EIV) regression [16]. EIV regression is implemented as the function eiv in the code on GitHub (https://github.com/mlekkha/LUCNORM).

#### 3.4.4. Robust Errors-in-Variable Regression

The normalized activity of the construct was determined as the slope of the line F=AR using robust errors-in-variable regression (REIV) [17]. Since the REIV algorithm is not available in any R package, we implemented it in R as follows.

*A* was estimated by minimizing the loss function
$$\sum_{i}\rho \left(\frac{{\left(1+A^{2}\right)}^{-\frac{1}{2}}\left(F_{i}-AR_{i}\right)}{S}\right),$$
where $R_{i}$ and $F_{i}$ are individual replicates of *Renilla* and firefly luminescence measurements, $\rho \left(t\right)=\frac{t^{2}}{6}\left(3-3\frac{t^{2}}{c^{2}}+\frac{t^{4}}{c^{4}}\right)$ is Tukey’s loss function with c=4.7, and *S* is an estimate of the scale of the residuals. The argument of Tukey’s function is the orthogonal distance of the point $\left(R_{i},F_{i}\right)$ from the regression line. Tukey’s function is bounded for large values of *t*, which limits the contribution of outliers to the loss function and ensures that the slope estimate is robust to outliers. The value of *S* was estimated by solving the equation
$$n^{-1}\sum \chi \left(\frac{{\left(1+A^{2}\right)}^{-\frac{1}{2}}\left(F_{i}-AR_{i}\right)}{S}\right)=\kappa ,$$
where κ=0.05 and χ(t) is Tukey’s loss function with c=1.56. The minimization problems were solved by the sequential least-squares quadratic programming (SLSQP) algorithm of the NLOPTR package of R, with parameters xtol_rel and maxeval set to $10^{-7}$ and 1000 respectively.

95% confidence intervals were estimated by bootstrapping using the R package BOOT. 999 replicates were subsampled using the ordinary simulation and the function boot.ci was used determine confidence intervals using the basic bootstrap method. REIV regression is implemented as the function robusteiv in the code on GitHub (https://github.com/mlekkha/LUCNORM).

### 3.5. Generation of Simulated Data

We generated simulated luminescence data that take into account sample-to-sample variation in transfection efficiency, random errors in both firefly and *Renilla* luminescence from other sources, and potential outliers. For each simulated experiment, we generated i=1,⋯,N samples. The transfection efficiency of each sample was drawn from the Beta distribution,
$$t_{i}∼B\left(\alpha ,\beta \right),$$
so that it varied between 0 and 1. The mean transfection efficiency of an experiment is $\bar{t}=\alpha /\left(\alpha +\beta \right)$. We simulated experiments with different mean transfection efficiencies (Figure 2) by varying α and β (Table 1).

The *Renilla* luminescence was computed by multiplying $t_{i}$ with the maximal *Renilla* luminescence *R* and introducing errors drawn from a Gaussian mixture model
(5)$$R_{i}=Rt_{i}+CN\left(\sigma_{11}^{2},\sigma_{12}^{2},0.05\right),$$
where $CN\left(\sigma_{11}^{2},\sigma_{12}^{2},\gamma \right)=\left(1-\gamma \right)N\left(0,\sigma_{11}^{2}\right)+\gamma N\left(0,\sigma_{12}^{2}\right)$ are random errors with variance $\sigma_{11}^{2}$ contaminated with errors with higher variance $\sigma_{12}^{2}$. The Gaussian mixture model simulates outliers occurring with a frequency of 5%.

Firefly luminescence is computed similarly,
$$F_{i}=ARt_{i}+CN\left(\sigma_{21}^{2},\sigma_{22}^{2},0.05\right),$$
with the maximal firefly luminescence given by the product of the “true” relative activity *A* and *R*. In the simulations presented here, we assumed that the standard deviations of firefly luminescence scale with the activity so that $\sigma_{21}=A\sigma_{11}$ and $\sigma_{22}=A\sigma_{12}$.

### 3.6. Procedure for Using REIV to Analyze Dual-Luciferase Reporter Data

We assume that a dual reporter experiment has been performed with an experimental design suitable for the biological question being investigated. We also assume that a sufficient number of replicate experiments have been performed with at least two experimental firefly Luciferase constructs and/or at least two experimental conditions.

We will utilize the example of the data underlying Figure 5 in subsequent description of the analysis procedure. In this example, three experimental constructs, *Cebpa(0)*, *Cebpa(7)*, and *Cebpa(7m1)*, were interrogated, which we will refer to as *0*, *7*, and *7m1* respectively for brevity. These constructs were assayed in PUER cells in two conditions, uninduced (no OHT) IL3 and induced (100 nM OHT) GCSF in 10 replicates per condition. Each experiment also included a reporter vector expressing *Renilla* luciferase under the control of the CMV promoter, and therefore resulted in two luminescence measurements, one for firefly Luciferase and the other for *Renilla* Luciferase.

#### 3.6.1. Installing Required R Packages

The following packages are required for using the REIV normalization code:bootparallelnloptrggplot2reshape2RColorBrewer

These may be installed either using the R Package Installer or by using the following command on the R Console:
install.packages("<package_name>")

#### 3.6.2. Downloading REIV Code from GitHub and Loading It into R

Download the code by visiting https://github.com/mlekkha/LUCNORM, clicking the Clone or download button and choosing Download ZIP.Uncompress the downloaded ZIP file.Set the working directory in R to the location of the downloaded code using Misc → Change Working Directory…Load the REIV analysis functions into the R workspace.
source("lucAnalysis.R")

source("robusteiv.R")


#### 3.6.3. Input Data Format

REIV R code accepts input data in a comma-separated values (CSV) file (see Appendix A for example) organized as follows (Table 2). The first and second columns are titled Luc and Ren respectively and contain the measured luminescences. The third column is titled Construct and contains the names of the firefly Luciferase constructs which were assayed. The rest of the columns contain the names of the conditions and are titled accordingly. In our example data (Appendix A), there are two additional columns titled OHT and Cytokine indicating whether the cells were induced with OHT or not and the name of the cytokine treatment respectively.

Once the CSV file has been prepared in this manner, the data are ready to be analyzed by REIV.

#### 3.6.4. Importing Input Data into the R Workspace and Preparing It

In the R Console, read the file into a variable.
data <- read.csv("<input_file_name>")
(Optional) If any of the construct/condition names are numeric, they must be set as categorical variables. This may be accomplished as follows.
data$<variable_name> <- as.factor(data$<variable_name>)
For example:
data$Construct <- as.factor(data$Construct)


#### 3.6.5. Normalizing and Saving Results

Once the data have been saved in a variable, REIV can be used to compute normalized Luciferase activity and its confidence interval for each combination of construct and condition.
 norm_activity <- calcSlopesCIs(<data_variable_name>,

               alpha = <alpha_value>,

               regmethod = robusteiv,

               cim = "boot_positive_ci",

               ignore=c("Luc", "Ren"))
Here, alpha is the significance threshold. For example, setting alpha to 0.05 will compute 95% confidence intervals. regmethod is the normalization method. Use regmethod = eiv for EIV normalization. cim is the method used for computing the confidence intervals. Use cim = "gleser_ci" for EIV normalization. ignore specifies which columns/variables are not conditions. If there are other columns in the CSV file that are not conditions, such as annotations, they should be included in the ignore vector.Example usage:
norm_activity <- calcSlopesCIs(data,

               alpha=0.05,

               regmethod=robusteiv,

               cim="boot_positive_ci",

               ignore=c("Luc", "Ren"))
The output of calcSlopesCIs is a data frame with columns containing the normalized Luciferase activity (slope) and the lower and upper confidence intervals (ci_lower and ci_upper). The values may be inspected with the print command.
print(norm_activity)
The normalized activities may be saved to a CSV file for further analysis or visualization using the write.csv function.
write.csv(<variable_name>, "<output_file_name>")
For example:
write.csv(norm_activity, "normalized_activities.csv")


## Figures and Tables

**Figure 1 mps-02-00062-f001:**
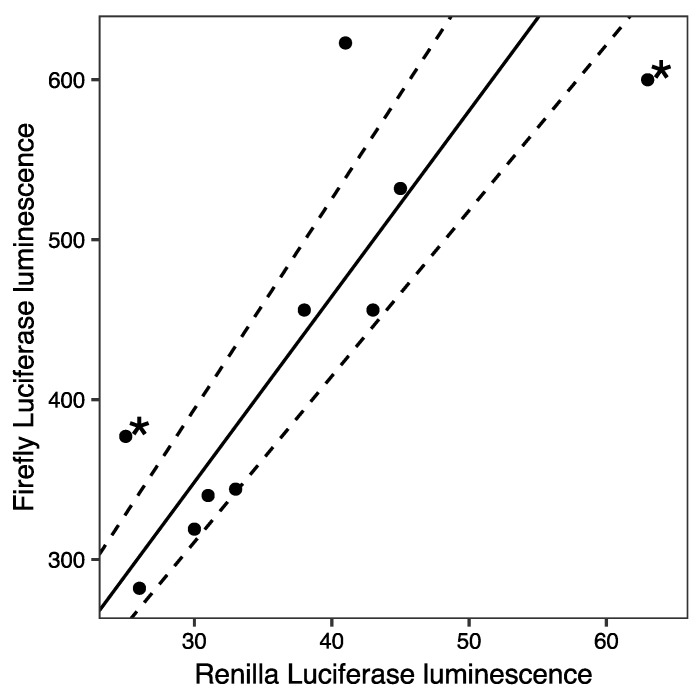
Example firefly and *Renilla* luminescence data from a myeloid cell line. Firefly *luciferase* was under the control of the *Cebpa* promoter, while *Renilla luciferase* was under the control of the CMV promoter. Luminescence is reported in relative luminescence units (RLUs). The best-fit line (solid) determined by robust errors-in-variable (REIV) regression is shown. Dashed lines represent the 95% confidence interval for slope determined by bootstrapping (Section 3.4.4). Potential outliers are indicated with asterisks.

**Figure 2 mps-02-00062-f002:**
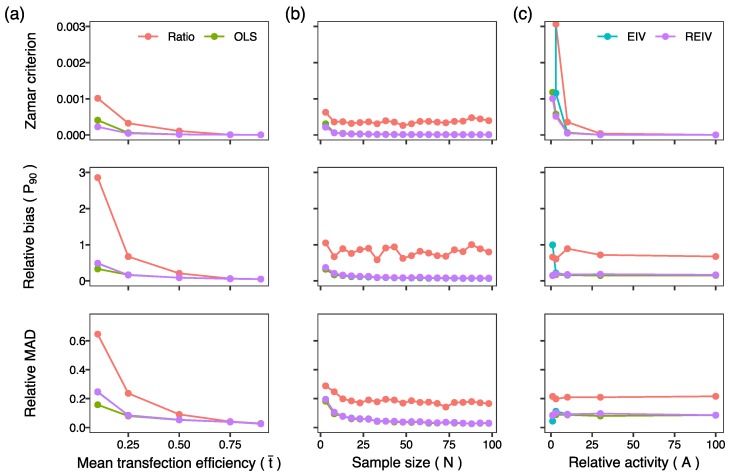
Tests of normalization methods on simulated data. The top panels plot the Zamar criterion, which is 0 for perfect inference of the true activity. The middle panels plot the 90th percentile ($P_{90}$) of the relative bias. The bottom panels plot the relative median absolute deviation, a measure of precision. (**a**) Performance of the methods plotted against mean transfection efficiency. True activity: A=10. Sample size: N=10. *Renilla* errors: $\sigma_{11}=\sigma_{12}=3$. Firefly errors were scaled with activity: $\sigma_{21}=\sigma_{22}=10\sigma_{11}$. (**b**) Performance plotted against sample size. Mean transfection efficiency: $\bar{t}=0.25$. The other parameters are the same as in panel (**a**,**c**). Performance plotted against true relative activity. Sample size: N=10. The other parameters are the same as in panel (**b**).

**Figure 3 mps-02-00062-f003:**
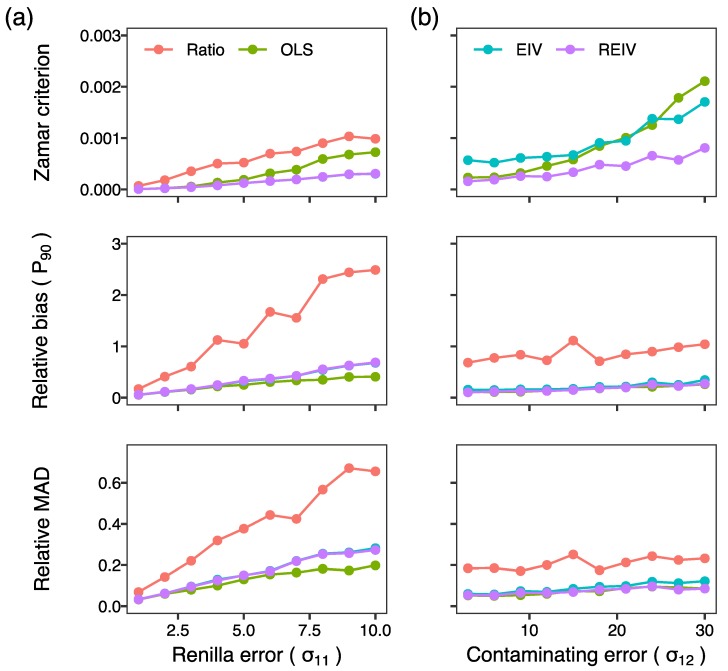
Tests of normalization methods on simulated data. The top panels plot the Zamar criterion, which is 0 for perfect inference of the true activity. The middle panels plot the 90th percentile ($P_{90}$) of the relative bias. The bottom panels plot the relative median absolute deviation, a measure of precision. (**a**) Performance of the methods plotted against standard deviation of the error in *Renilla* luminescence. Mean transfection efficiency: $\bar{t}=0.25$. True activity: A=10. Sample size: N=10. There were no outliers: $\sigma_{12}=\sigma_{11}$. Firefly errors were scaled with activity: $\sigma_{21}=\sigma_{22}=10\sigma_{11}$ (**b**). Performance plotted against increasing severity of outliers. The standard deviation of the contaminating *Renilla* errors $\sigma_{12}$ was varied to simulate outliers. True activity: A=3. *Renilla* errors: $\sigma_{11}=3$. Firefly errors: $\sigma_{21}=9$. The standard deviation of firefly contaminating errors was scaled with activity: $\sigma_{22}=3\sigma_{12}$. The other parameters are the same as in panel (**a**). The Zamar criterion of the ratiometric method was greater than the upper limit of the *y*-axis.

**Figure 4 mps-02-00062-f004:**
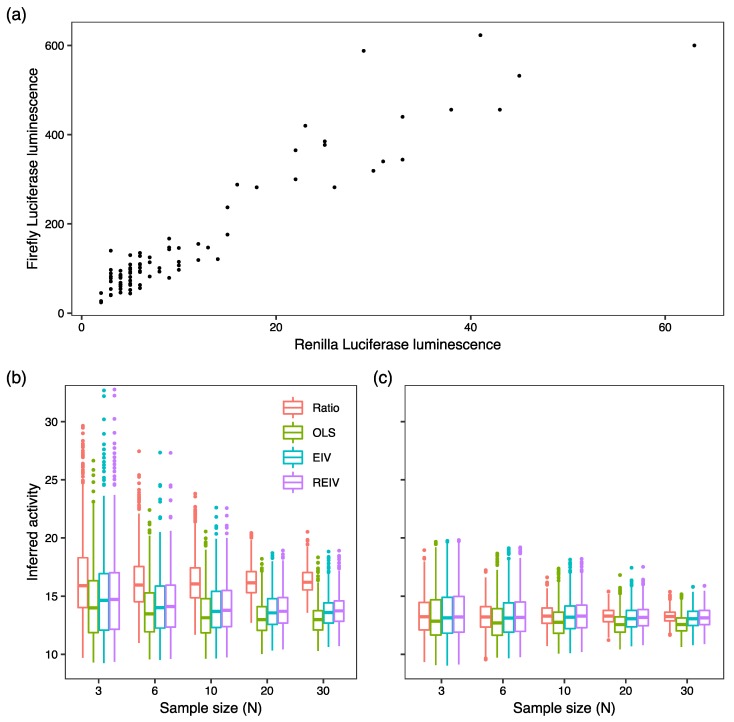
Tests of normalization methods on empirical data. (**a**) Dataset of firefly luminescence driven by the *Cebpa* promoter and *Renilla* luminescence driven by the CMV promoter in PUER cells. N=85. The dataset was sampled 1000 times with varying sample sizes. Promoter activity was estimated with each method from the same exact data. (**b**) Boxplots of the inferred activities. The box lines are the first quartile, median, and the third quartile. The whiskers extend to the most extreme values lying within 1.5 times the interquartile range, and any datapoints outside the whiskers are shown as circles. (**c**) Datapoints having *Renilla* luminescence 8 RLU or less were excluded from the analysis.

**Figure 5 mps-02-00062-f005:**
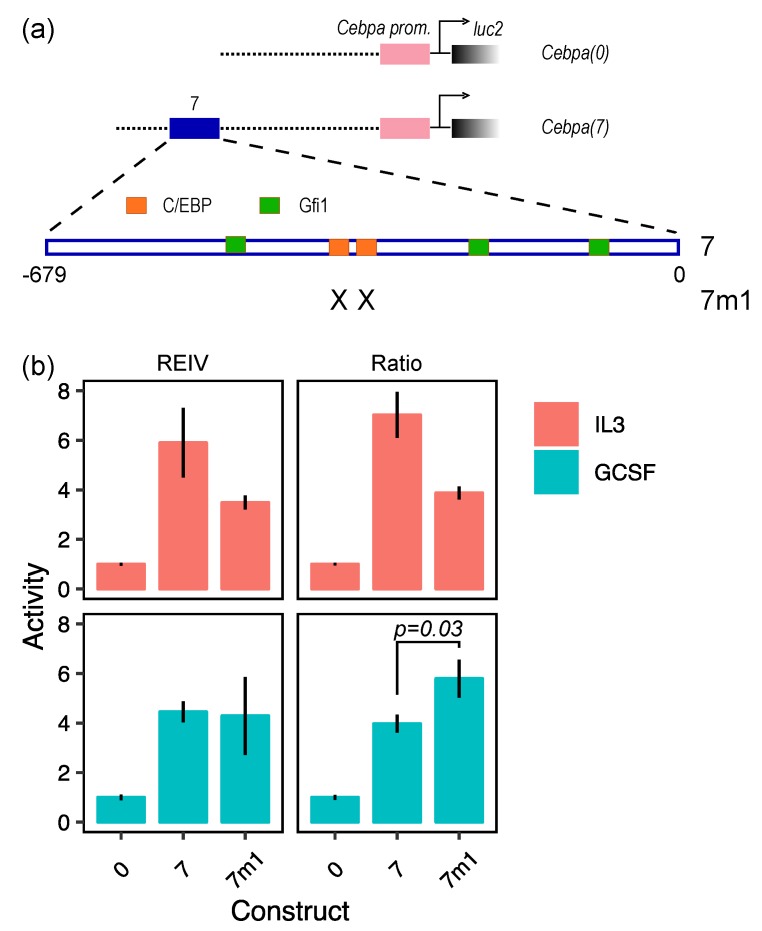
Performance of normalization methods in detecting the effect of mutations on enhancer activity. (**a**) Design of reporter construct. *Cebpa(0)* contains the *luc2* gene driven by the proximal *Cebpa* promoter. *Cebpa(7)* contains an enhancer in addition to the proximal promoter [4]. Binding sites for C/EBP family transcription factors and Gfi1 are shown in the magnified view. C/EBP sites have been mutated in *Cebpa(7m1)*. (**b**) The relative activity of *Cebpa(0)*, *Cebpa(7)*, and *Cebpa(7m1)* inferred by either the REIV (left panels) or the ratiometric (right panels) methods in uninduced IL3 (progenitor) or induced GCSF (neutrophil) conditions. The activity has been normalized against the activity of the proximal promoter (*Cebpa(0)*). Error bars are standard errors of the mean. The ratiometric method spuriously detects an activation of *Cebpa(7m1)* in GCSF conditions (bottom right panel; Welch two sample *t*-test, N=10, p=0.03).

**Table 1 mps-02-00062-t001:** Values of Beta distribution parameters, α and β, for simulating different mean transfection efficiencies ($\bar{t}$).

$\bar{t}$	α	β
0.1	2	18
0.25	2	6
0.5	2	2
0.75	6	2
0.9	18	2

**Table 2 mps-02-00062-t002:** Format of the comma-separated values (CSV) file containing luminescence data.

Luc	Ren	Construct	Condition 1	Condition 2	⋯
Luc luminescence	Ren luminescence	Name	Name	Name	⋯
⋮	⋮	⋮	⋮	⋮	⋮

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
