# Peer review of "Robust Normalization of Luciferase Reporter Data"

_mps, 2019, doi:10.3390/mps2030062_

Round 1

Reviewer 1 Report

The method will be valuable to biologists who perform steady state luciferase assays especially in cells that have low transfection efficiency or are difficult to transfect. The reviewer is not qualified to comment on the math, but I have concerns that require the authors' attention. The method description may be improved to make it easier to follow and more intuitive for biologists. Based on the data in Fig. 4B-C, it is not obvious how REIV is better than OLS and EIV and whether it is significantly better than ratiometric. How do these methods lead to different conclusions? The ratiometric method deviates from the other three methods, but it is not clear how this might impact the conclusion. It would help to illustrate the point if the author can provide an example in which the method used does matter. The experimental data presented in Fig. 4 have very different levels of renilla activity proportional to firefly, which argues against confounding effects of low transfection efficiency. Finally, it would be helpful to give better explanation for biologists on how to use the algorithm. In summary, the authors need to explain how REIV is the method of choice (significantly better than others) and whether it contributes to analysis that is of biological significance. 

Reviewer 2 Report

In this work by Repele and Manu, four methods (ratiometric, OLS, EIV, and REIV) were evaluated to establish the best way to normalize for experimental errors in transient reporter assays, using simulated data or empirical measurements of Cebpa promoter activity in PUER cells.

In particular authors found that the ratiometric method to normalize luciferase activity, performs poorly in cells with low transfection efficiency generating data with a high coefficient of variation.

In the end the authors show that the errors-in-variables (REIV) regression method performs the best in normalizing luciferase reporter data.

Finding a way to properly normalize data from reporter gene assays is crucial to produce reliable interpretable results from this very powerful investigational method, therefore this is a valuable work.

Nevertheless, one important issue must be solved before publication, and that is the fact that the authors neglected to consider that one of the biggest bias related to reporter gene assay occurs when cells are stimulated with a cytokine, a drug, or anything that can, not only influence the main promoter and/or the enhancer driving the reporter gene product accumulation, but also the transcription of the reporter gene used for normalization (i.e. Renilla luciferase) in a non-specific manner.

Moreover, in order to minimize such non-specific effects a promoter different from the CMV promoter, which is highly influenced by cellular stimulation, should be used, like for example a null promoter containing Renilla luciferase gene.

Therefore, the authors should perform the same empirical measurements using a null promoter containing Renilla luciferase gene, and also in the presence of a stimulus known to activate the main luciferase reporter construct, in order to validate the errors-in-variables (REIV) regression method for normalization.

Round 2

Reviewer 1 Report

The authors have addressed all my concerns in the revison. 

Reviewer 2 Report

Authors performed the requested modifications and experiments, therefore the manuscript is now suitable for pubblication.